# PeerJ

# Fluvial transport potential of shed and root-bearing dinosaur teeth from the late Jurassic Morrison Formation

Joseph E. Peterson[1], Jason J. Coenen[2] and Christopher R. Noto[3]

[1] Department of Geology, University of Wisconsin-Oshkosh, Oshkosh, WI, United States
[2] Department of Geology and Environmental Geosciences, Northern Illinois University, DeKalb, IL, United States
[3] Department of Biological Sciences, University of Wisconsin-Parkside, Kenosha, WI, United States

Corresponding author
Joseph E. Peterson,
petersoj@uwosh.edu

## ABSTRACT

Shed dinosaur teeth are commonly collected microvertebrate remains that have been used for interpretations of dinosaur feeding behaviors, paleoecology, and population studies. However, such interpretations may be biased by taphonomic processes such as fluvial sorting influenced by tooth shape: shed teeth, removed from the skull during life, and teeth possessing roots, removed from the skull after death. As such, teeth may behave differently in fluvial systems due to their differences in shape. In order to determine the influence of fluvial processes on the preservation and distribution of shed and root-bearing dinosaur teeth, the hydrodynamic behaviors of high-density urethane resin casts of shed and root-bearing *Allosaurus* and *Camarasaurus* teeth were experimentally tested for relative transport distances at increasing flow velocities in an artificial fluviatile environment. Results show that tooth cast specimens exhibited comparable patterns of transport at lower velocities, though the shed *Camarasaurus* teeth transported considerably farther in medium to higher flow velocities. Two-Way ANOVA tests indicate significant differences in the mean transport distances of tooth casts oriented perpendicular to flow ($p < 0.05$) with varying tooth morphologies and flow velocities. The differences exhibited in the transportability of shed and root-bearing teeth has important implications for taphonomic reconstructions, as well as future studies on dinosaur population dynamics, paleoecology, and feeding behaviors.

## INTRODUCTION

Experiments on the transport of skeletal remains in controlled fluvial systems have been of significant use in deciphering relative hydrodynamic properties and behaviors of remains in vertebrate taphonomic studies (e.g., *Voorhies, 1969*; *Behrensmeyer, 1975*; *Boaz & Behrensmeyer, 1976*; *Hanson, 1980*; *Blob, 1997*; *Nasti, 2005*; *Peterson & Bigalke, 2013*). A majority of previous flume experiments have been conducted on a variety of macrovertebrate taxonomic groups, such as mammals and dinosaurs (e.g., *Voorhies, 1969*; *Behrensmeyer, 1975*; *Boaz & Behrensmeyer, 1976*; *Coard & Dennell, 1995*; *Coard, 1999*;

*Nasti, 2005*; *Peterson & Bigalke, 2013*). Although microvertebrate remains are commonly collected and utilized for paleoecological and taphonomic reconstructions, few studies have employed flume experiments to explore the role of differing relative hydrodynamic properties in the development of microvertebrate assemblages or "microsites" (e.g., *Dodson, 1973*; *Blob, 1997*; *Trapani, 1998*).

"Microsites" are accumulations of small, fragmentary, moderately to well-sorted fossil material, including largely disarticulated vertebrate remains, typically dominated by fish scales, bone fragments, and shed teeth (*Wood, Thomas & Visser, 1988*). Although scales and bone fragments are of interest for their potential uses in taphonomic reconstructions (e.g., *Blob & Fiorillo, 1996*; *Wilson, 2008*; *Peterson, Scherer & Huffman, 2011*), the abundance of shed dinosaur teeth in Mesozoic deposits is of particular interest in attempts to infer dental physiology (*Sereno & Wilson, 2005*; *D'Emic et al., 2013*), feeding behaviors (*Jennings & Hasiotis, 2006*), paleoecology (*Bakker & Bir, 2004*), and their potential for population studies (*Erickson, 1996*).

However, interpretations regarding feeding behaviors, paleoecology, and population dynamics based on shed teeth may be biased by taphonomic processes such as fluvial sorting influenced by tooth shape: shed teeth (removed from the skull *in vivo*) and teeth possessing roots (removed from the skull *post-mortem*) may behave differently in fluvial settings due to their shape differences. In order to determine the role of fluvial processes on the preservation and distribution of shed and root-bearing dinosaur teeth, an experiment was conducted to ascertain the hydrodynamic properties of two morphologically distinct sets of dinosaur teeth from Late Jurassic theropods and sauropods. Specifically, the question is addressed: Are the mean transport distances the same for shed and root-bearing teeth at varying flow velocities? Presented here are the results of this experiment and a discussion on the potential biases of shed teeth in the fossil record.

## MATERIALS AND METHODS

To test for variation in relative transport distances in theropod and sauropod teeth in fluvial settings, casts were made of four different dinosaur teeth using a urethane resin and placed in a recirculating flume at increasing stages of flow velocity. Casts were chosen instead of fossil teeth in order to avoid damage to delicate fossil specimens, and to maintain a consistent specific gravity among specimens. Tooth casts were produced using Replicator 400[TM](Alumilite), which has a cured specific gravity of approximately 1.5 g/cm$^3$. Enamel and dentine have specific gravities of 2.8 g/cm$^3$ and 2.3 g/cm$^3$, respectively (*Brekhus & Armstrong, 1935*). While the specific gravity of the casting resin is different than that of teeth, relative comparisons can be conducted among cast elements of different shapes with the use of this standardized specific gravity.

The four specimens of dinosaur teeth were chosen based on their differences in shape, size, and representation in the fossil record (*Blob & Fiorillo, 1996*) (Table 1). To model theropod and sauropod teeth associated with *post-mortem* cranial disarticulation, a single set of casts was produced of root-bearing maxillary tooth specimens of *Camarasaurus* (UWO-VPC-2013.003) and *Allosaurus* (UWO-VPC-2013.001) (Figs. 1A and 1B). The

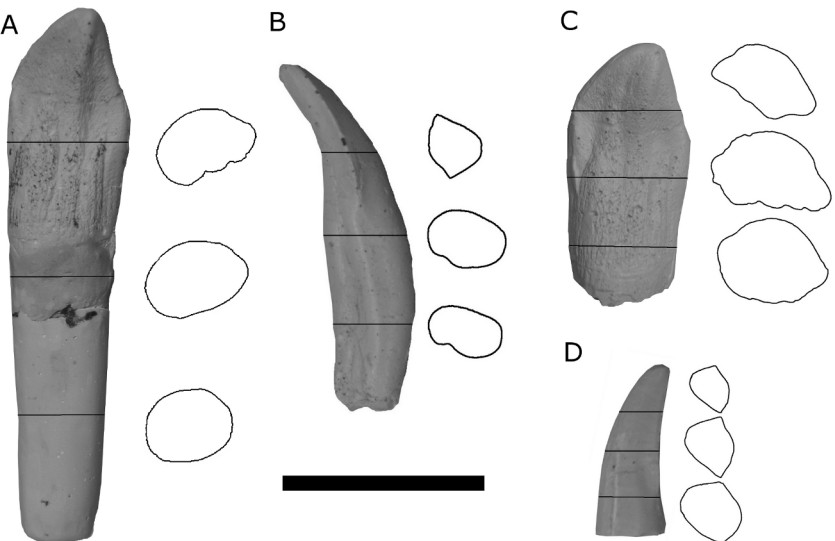

**Figure 1 Photographs and cross-sectional outlines of cast specimens used in the flume experiment.** (A) Root-bearing *Camarasaurus* tooth (UWO-VPC-2013.003), (B) Root-bearing *Allosaurus* tooth (UWO-VPC-2013.001), (C) Shed *Camarasaurus* tooth (UWO-VPC-2013.004), and (D) Shed *Allosaurus* tooth (UWO-VPC-2013.002). Scale bar = 5 cm.

**Table 1 Dimensions and properties of cast tooth specimens.**

| Tooth specimen | Mass of cast (g) | Length (mm) | Width (mm) | Density (g/cm$^3$) | Shape |
|---|---|---|---|---|---|
| *Camarasaurus* (shed) | 29 | 61 | 29 | 1.45 | Compact |
| *Camarasaurus* (rooted) | 60 | 122 | 29 | 1.5 | Elongate |
| *Allosaurus* (shed) | 8.2 | 44 | 19 | 1.49 | Conical |
| *Allosaurus* (rooted) | 19 | 82 | 19 | 1.52 | Elongate |

*Camarasaurus* tooth cast UWO-VPC-2013.003 was made from a shed crown and attached to a sculpted root. Similarly, to model shed theropod and sauropod teeth associated with tooth regeneration *in vivo*, a second set of casts were produced (UWO-VPC-2013.002 and UWO-VPC-2013.004) with the root portions of the casts removed (Figs. 1C and 1D). By using the same tooth crowns of *Allosaurus* and *Camarasaurus* and secondarily adding or removing the root portions of the casts for the experiment, more control over the role of attached roots in transport could be observed. The casts used in this study are housed at the University of Wisconsin Oshkosh Department of Geology, and were based on specimens in private collections. Casts were also digitized into 3D models using a NextEngine Desktop 3D Scanner and processed with ScanStudio HD Pro (NextEngine) (Figs. S1–S4, Text S1).

Transport experiments were conducted at the re-circulating flume at the University of Wisconsin-Oshkosh Department of Geology. The flume measures 0.45 m deep × 0.15 m wide and 3.5 m in length (Fig. 2), and was filled to maintain a depth of 10 cm during trials. To determine relative transport distances associated with flow velocity, tests were conducted on a planar glass surface in 10 cm water depth. Each tooth cast was repeatedly

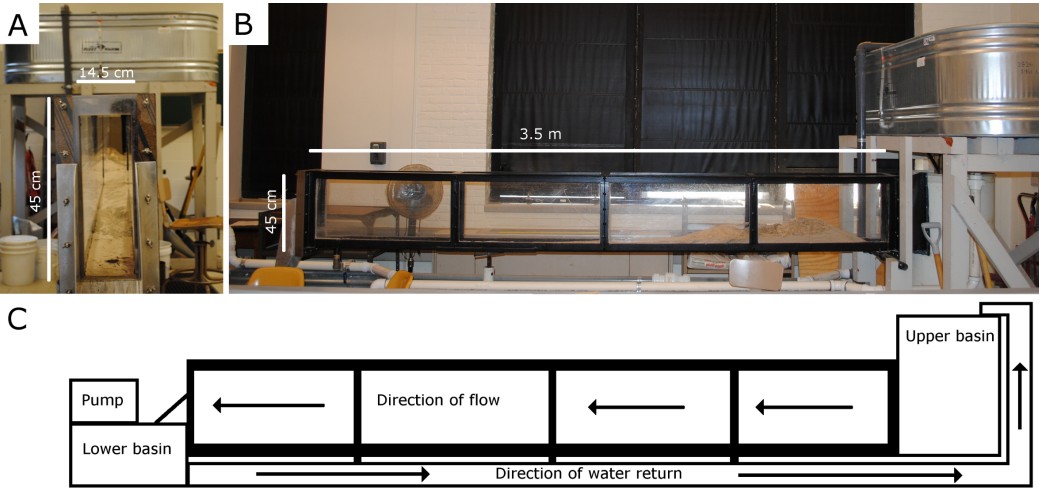

**Figure 2 Recirculating flume facility at UW Oshkosh where experiments were conducted.** (A) Cross-sectional view, (B) Side view, (C) Schematic diagram of recirculating flume.

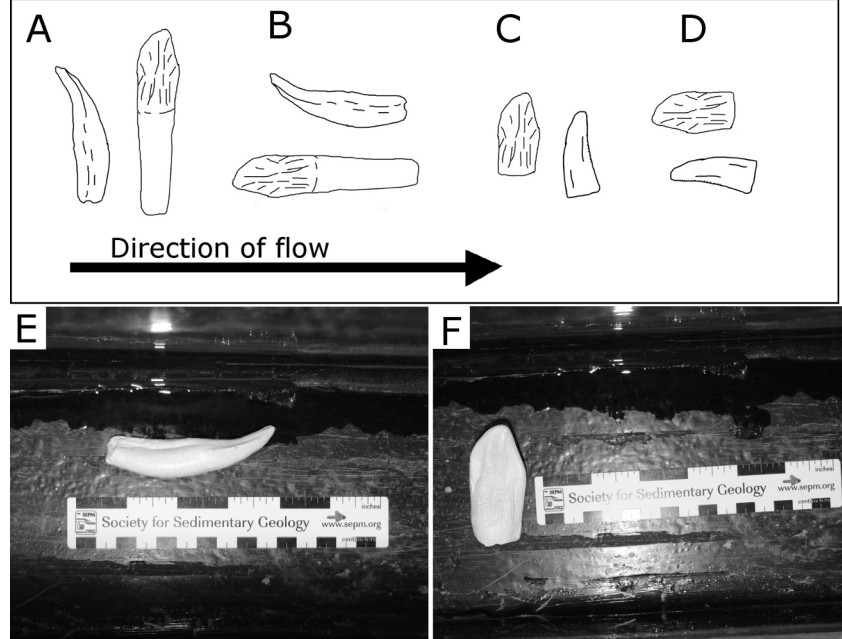

**Figure 3 Examples of orientations of tooth casts.** (A) Root-bearing casts oriented perpendicular to flow, (B) root-bearing casts oriented parallel to flow, (C) shed casts oriented perpendicular to flow, (D) shed casts oriented parallel to flow. (E) Example of root-bearing *Allosaurus* tooth cast oriented parallel to flow, (F) example of shed *Camarasaurus* tooth oriented perpendicular to flow.

placed in the flume perpendicular and parallel to flow (Figs. 3A–3F) at three different velocity settings; 10.0–19.9 cm/s, 20.0–29.9 cm/s, and 30.0–39.9 cm/s. The apex of the tooth crown was pointed in the upstream direction for trials where the teeth were placed parallel to flow (Figs. 3B and 3D). Additionally, trials ran in the perpendicular direction involved placing the apex of the tooth crown perpendicular to flow (Figs. 3A and 3C). Each

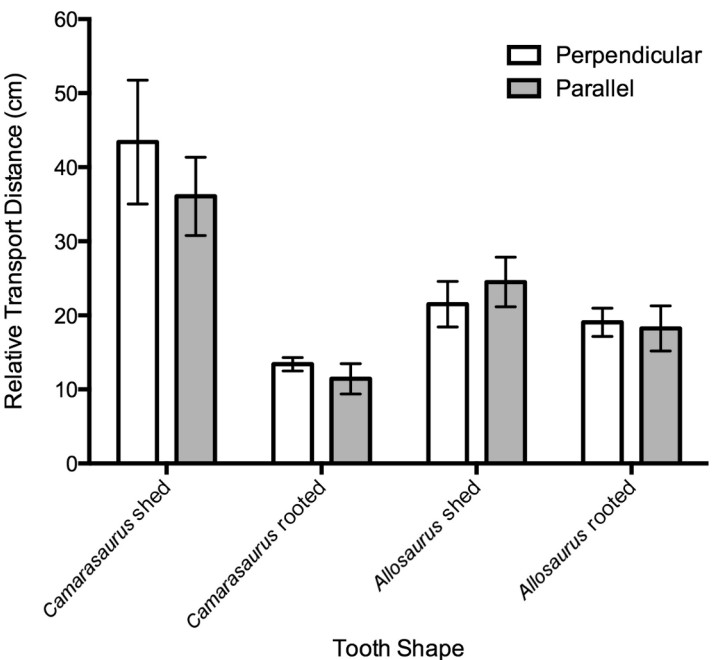

**Figure 4  Bar chart of transport distances for tooth casts.** Placed (A) perpendicular or (B) parallel to flow. Error bars represent standard error.

test consisted of 10 trials per tooth cast in each orientation and at each velocity stage. To avoid interactions between tooth casts during transport, casts were placed in the flume alone for the duration of the experiment. Total transport distance and flow velocity at the location of settling were collected for each trial. Relative transport distance serves as a proxy for relative time of transport and offers insight into time averaging (*Aslan & Behrensmeyer, 1996*). Relative transport distance data also serve as comparisons of the relative transportability among tooth casts. Entrainment velocity, the velocity required to move the casts, was determined by recording the fluvial velocity (HACH FH950 Portable Velocity System) at the location of settling.

## Statistical methods

Analysis of variance (ANOVA) was employed to compare the mean transport distances of the four tooth casts under different flow velocities. A D'Agustino and Shapiro omnibus test found some data departed from a Gaussian distribution (normality), therefore the data were log-transformed prior to analysis (*Sokal & Rohlf, 1995*). An initial one-way ANOVA found no significant difference between the transport distances of parallel- and perpendicular-oriented datasets (Fig. 4); each dataset is therefore analyzed independently. A two-factor ANOVA followed by a Bonferroni multiple comparisons test was run for each dataset. The Bonferroni test compares the simple effects of tooth cast shape within each velocity range, utilizing a conservative single-family grouping for all comparisons. A nominal significance level of 0.05 was used in all ANOVA tests to reject the null hypothesis that the mean transport distances are the same for all tooth shapes and at all flow velocities.
All analyses were carried out using Prism version 6.0d for Macintosh (GraphPad Software, La Jolla, CA, USA, www.graphpad.com).

## RESULTS

During flume tests, teeth commonly initiated transport by sliding on the bottom of the flume; however, in a few instances, teeth rolled for a short distance and then slid to their final position. Two-factor ANOVA tests produced multiple significant results. Perpendicular tooth casts were found to vary significantly in transport distance due to tooth morphology ($F = 14.9$, $df = 3$, $p < 0.0001$) and flow velocity ($F = 54.35$, $df = 2$, $p < 0.0001$), with a strong interaction effect ($F = 4.865$, $df = 6$, $p = 0.0002$) (Table 2A). The strong interaction effect is explained by the Bonferroni test results, which indicates that significant differences occur between the shed *Camarasaurus* tooth and all other tooth cast specimens, mainly at high velocities (30–39.9 cm/s) (Table 2B). The variability is further enhanced by some other significant differences between teeth in the other flow velocity ranges.

Two-factor ANOVA results of tooth casts oriented parallel to flow indicated significant differences in mean transport distance due to tooth morphology ($F = 42.80$, $df = 3$, $p < 0.0001$) and flow velocity ($F = 78.45$, $df = 2$, $p < 0.0001$) (Table 2C), with a significant interaction effect ($F = 3.507$, $df = 6$, $p = 0.0033$). The Bonferroni test shows that several comparisons are significantly different across flow velocity ranges; occurring mainly between the shed and rooted teeth (Table 2D).

The initial orientation of the tooth (parallel vs. perpendicular) had no significant effect on relative transport distance (Fig. 4). The most notable difference in hydrodynamic behavior is observed between shed and rooted teeth, where shed teeth on average travelled further than rooted teeth under most conditions (Figs. 5A and 5B). However, the hydrodynamic behavior of each tooth shape varies with flow velocity. At lower flow velocities the teeth behave more similarly to each other, diverging significantly at higher flow velocities. This has been previously noted for other skeletal elements during fluvial transport (e.g., *Voorhies, 1969*).

## DISCUSSION

These results demonstrate a close link between shape differences in vertebrate teeth and their potential representation in a fossil assemblage due to the influence of shape on hydrodynamic behavior (*Behrensmeyer, 1975*; *Coard & Dennell, 1995*; *Peterson & Bigalke, 2013*).

Shed and root-bearing teeth differ significantly in hydrodynamic behavior and thus have an increased likelihood of contributing preservational biases; elongate teeth (i.e., root-bearing) and teeth approaching a conical shape (i.e., shed theropod teeth) do not transport as far with increasing flow velocities as compact teeth (i.e., shed *Camarasaurus* teeth). This suggests that compact teeth have a higher potential for continued transport while elongate and conical teeth are more likely to remain as lag, thus increasing their potential for preservation in the fossil record.

This suggestion may be tested by comparing the abundance, taphonomic signatures (i.e., quartz-grain equivalence, sorting, weathering, etc.), and proximity of root-bearing

**Table 2  Statistical tests performed on transport data.** Two-factor ANOVA (A) and Bonferroni multiple comparison test (B) results for tooth cast transport distances tested perpendicular to flow; two-factor ANOVA (C) and Bonferroni multiple comparison test (D) results for tooth cast transport distances tested parallel to flow. Adjusted $P$ value refers to the exact multiplicity-adjusted $p$-value calculated in Prism version 6.0d. All values based on log-transformed data.

**(A) Two-factor ANOVA table for perpendicular**

| Source of variation | SS | DF | MS | F | P value |
|---|---|---|---|---|---|
| Interaction | 1.367 | 6 | 0.2278 | 4.87 | $P = 0.0002$ |
| Flow velocity | 5.091 | 2 | 2.546 | 54.35 | $P < 0.0001$ |
| Tooth type | 2.093 | 3 | 0.6977 | 14.90 | $P < 0.0001$ |
| Residual | 5.058 | 108 | 0.04683 | | |

**(B) Two-factor ANOVA table for parallel**

| Source of variation | SS | DF | MS | F | P value |
|---|---|---|---|---|---|
| Interaction | 0.9006 | 6 | 0.1501 | 3.507 | $P = 0.0033$ |
| Flow velocity | 6.715 | 2 | 3.357 | 78.45 | $P < 0.0001$ |
| Tooth type | 5.495 | 3 | 1.832 | 42.80 | $P < 0.0001$ |
| Residual | 4.622 | 108 | 0.04280 | | |

**(C) Bonferroni Multiple Comparison Test results for perpendicular**

| Comparison | Mean diff. | 95% CI of diff. | t | DF | Adjusted P value |
|---|---|---|---|---|---|
| Low flow (10–19.9 cm/s) | | | | | |
| Camarasaur shed vs. Allosaur shed | 0.2682 | 0.008091 to 0.5283 | 2.771 | 108 | 0.0395 |
| Camarasaur shed vs. Allosaur rooted | 0.2569 | −0.00324 to 0.5170 | 2.654 | 108 | 0.0549 |
| Camarasaur shed vs. Camarasaur rooted | 0.09858 | −0.1615 to 0.3587 | 1.019 | 108 | >0.9999 |
| Allosaur shed vs. Allosaur rooted | −0.01133 | −0.2715 to 0.2488 | 0.1171 | 108 | >0.9999 |
| Allosaur shed vs. Camarasaur rooted | −0.1696 | −0.4298 to 0.09048 | 1.753 | 108 | 0.4949 |
| Allosaur rooted vs. Camarasaur rooted | −0.1583 | −0.4184 to 0.1018 | 1.636 | 108 | 0.6289 |
| Intermediate flow (20–29.9 cm/s) | | | | | |
| Camarasaur shed vs. Allosaur shed | 0.07721 | −0.1829 to 0.3373 | 0.7977 | 108 | >0.9999 |
| Camarasaur shed vs. Allosaur rooted | 0.03548 | −0.2246 to 0.2956 | 0.3666 | 108 | >0.9999 |
| Camarasaur shed vs. Camarasaur rooted | 0.3358 | 0.07571 to 0.5960 | 3.470 | 108 | 0.0045 |
| Allosaur shed vs. Allosaur rooted | −0.04173 | −0.3019 to 0.2184 | 0.4312 | 108 | >0.9999 |
| Allosaur shed vs. Camarasaur rooted | 0.2586 | −0.00149 to 0.5187 | 2.672 | 108 | 0.0522 |
| Allosaur rooted vs. Camarasaur rooted | 0.3004 | 0.04024 to 0.5605 | 3.103 | 108 | 0.0147 |
| High flow (30–39.9 cm/s) | | | | | |
| Camarasaur shed vs. Allosaur shed | 0.3733 | 0.1131 to 0.6334 | 3.857 | 108 | 0.0012 |
| Camarasaur shed vs. Allosaur rooted | 0.4872 | 0.2271 to 0.7473 | 5.034 | 108 | <0.0001 |
| Camarasaur shed vs. Camarasaur rooted | 0.6446 | 0.3845 to 0.9047 | 6.660 | 108 | <0.0001 |
| Allosaur shed vs. Allosaur rooted | 0.1140 | −0.1462 to 0.3741 | 1.177 | 108 | >0.9999 |
| Allosaur shed vs. Camarasaur rooted | 0.2713 | 0.01123 to 0.5315 | 2.804 | 108 | 0.0359 |
| Allosaur rooted vs. Camarasaur rooted | 0.1574 | −0.1027 to 0.4175 | 1.626 | 108 | 0.6409 |

| (D) Bonferroni Multiple Comparison Test results for parallel | | | | | |
|---|---|---|---|---|---|
| Comparison | Mean diff. | 95% CI of diff. | $t$ | DF | Adjusted $P$ value |
| Low flow (10–19.9 cm/s) | | | | | |
| Camarasaur shed vs. Allosaur shed | 0.08950 | −0.1592 to 0.3382 | 0.9675 | 108 | >0.9999 |
| Camarasaur shed vs. Allosaur rooted | 0.3681 | 0.1195 to 0.6168 | 3.979 | 108 | 0.0008 |
| Camarasaur shed vs. Camarasaur rooted | 0.5345 | 0.2858 to 0.7831 | 5.777 | 108 | <0.0001 |
| Allosaur shed vs. Allosaur rooted | 0.2786 | 0.02995 to 0.5273 | 3.011 | 108 | 0.0194 |
| Allosaur shed vs. Camarasaur rooted | 0.4450 | 0.1963 to 0.6936 | 4.810 | 108 | <0.0001 |
| Allosaur rooted vs. Camarasaur rooted | 0.1664 | −0.08231 to 0.4150 | 1.798 | 108 | 0.4497 |
| Intermediate flow (20–29.9 cm/s) | | | | | |
| Camarasaur shed vs. Allosaur shed | 0.1847 | −0.06400 to 0.4333 | 1.996 | 108 | 0.2908 |
| Camarasaur shed vs. Allosaur rooted | 0.3013 | 0.05265 to 0.5500 | 3.257 | 108 | 0.0090 |
| Camarasaur shed vs. Camarasaur rooted | 0.8272 | 0.5785 to 1.076 | 8.941 | 108 | <0.0001 |
| Allosaur shed vs. Allosaur rooted | 0.1167 | −0.1320 to 0.3653 | 1.261 | 108 | >0.9999 |
| Allosaur shed vs. Camarasaur rooted | 0.6425 | 0.3939 to 0.8912 | 6.945 | 108 | <0.0001 |
| Allosaur rooted vs. Camarasaur rooted | 0.5259 | 0.2772 to 0.7745 | 5.684 | 108 | <0.0001 |
| High Flow (30–39.9 cm/s) | | | | | |
| Camarasaur shed vs. Allosaur shed | 0.1819 | −0.06672 to 0.4306 | 1.967 | 108 | 0.3108 |
| Camarasaur shed vs. Allosaur rooted | 0.3071 | 0.05845 to 0.5558 | 3.320 | 108 | 0.0074 |
| Camarasaur shed vs. Camarasaur rooted | 0.3655 | 0.1168 to 0.6141 | 3.950 | 108 | 0.0008 |
| Allosaur shed vs. Allosaur rooted | 0.1252 | −0.1235 to 0.3738 | 1.353 | 108 | >0.9999 |
| Allosaur shed vs. Camarasaur rooted | 0.1835 | −0.06512 to 0.4322 | 1.984 | 108 | 0.2989 |
| Allosaur rooted vs. Camarasaur rooted | 0.05837 | −0.1903 to 0.3070 | 0.6309 | 108 | >0.9999 |

teeth to their original cranial elements. Indeed, root-bearing teeth are typically discovered relatively close to other skeletal remains, as they were removed during *post-mortem* cranial disarticulation and show relatively little transport (e.g., *Breithaupt, 2001*; *Lehman & Coulson, 2002*; *Derstler & Myers, 2008*). This is also supported by the high frequency of shed theropod teeth associated with proposed feeding sites (e.g., *Argast et al., 1987*; *Bakker, 1997*; *Jennings & Hasiotis, 2006*; *Roach & Brinkman, 2007*). However, the presence of both root-bearing and shed teeth of a variety of Jurassic dinosaurs in multi-taxa bonebeds (e.g., Cleveland Lloyd Dinosaur Quarry, Nail Quarry, and Quarry 9) may indicate a more complex origin of such assemblages, and require an understanding of the hydrodynamic properties of both root-bearing and shed teeth. Assessing the complexity of a tooth assemblage could be accomplished by measuring the relative proportions of each tooth shape (compact, conical, and elongate) present in the sample. Excessive numbers of compact teeth, for example, could be evidence of a heavily reworked or allochthonous origin of teeth in the assemblage.

These results provide further support for an interaction between conditions in the depositional environment and transported elements. Despite finding no statistically significant difference in average transport distance between perpendicular and parallel orientations, a great degree of variability occurred within and between the different velocity ranges.

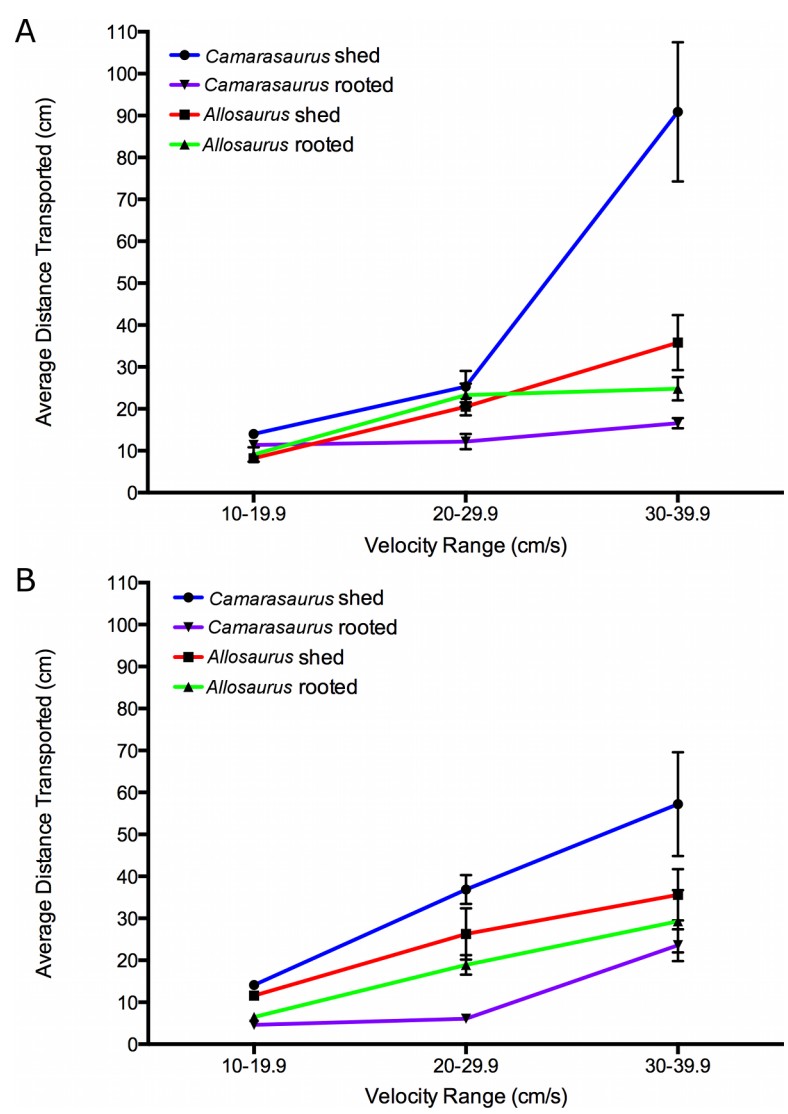

**Figure 5  Average transport distance of cast tooth specimens versus velocity ranges.** (A) Specimens tested perpendicular to flow and (B) parallel to flow. Error bars represent standard error.

Hydrodynamic behavior (as measured by relative transport distance) depended on the flow velocity. If flow conditions were not interacting with the hydrodynamic properties of each tooth, one would expect to see a linear response and relatively fixed differences among all tooth morphologies. Non-linear responses were found across all velocity ranges and tooth morphologies. This variability is most apparent in perpendicular-oriented trials. Environment of deposition plays a role in assembly of lags and microsites (*Rogers & Brady, 2010*). More kinds of teeth of varying shapes may indicate shorter transport distance, whereas many similar kinds of teeth may be more affected by transport (either carried in or winnowed). It is important to consider that differences in substrate could have implications not addressed in this preliminary study of hydrodynamic tooth behavior.

Further work exploring these questions, including interactions with different substrate types, will be necessary.

The variable transportability of shed and root-bearing teeth has important implications for taphonomic reconstructions. For example, shed theropod and sauropod teeth in Morrison quarries and microsites have previously been used in paleoecological reconstructions (*Bakker & Bir, 2004*), inferences of feeding behaviors (*Jennings & Hasiotis, 2006*), and suggested for utilization in population studies (*Erickson, 1996*). However, prior to such inferences, hydrodynamic properties, such as relative transport potential, must be taken into consideration in order to gain a better understanding of whether the presence of shed teeth represent an allochthonous or autochthonous component to such quarries.

The results shown here indicate that, not only does tooth morphology matter in transport potential, but the interaction between hydrodynamic properties of tooth shape and conditions in the depositional environment that contribute to microfossil accumulations. These results suggest that tooth shape and flow velocity interact to influence transport, but the hydrodynamic behavior of teeth becomes increasingly unpredictable at higher flow velocities, and therefore may limit our ability to infer the taphonomic history of a microfossil assemblage from high-velocity depositional environments. Although this study focused on just two common Jurassic taxa, further experimental studies on the potential transportability of shed teeth of varying morphologies and substrate conditions have the potential to indicate preservation biases in bonebed and microfossil assemblages. Understanding the potential of such biases may influence further interpretations of dinosaur population dynamics, paleoecology, and feeding behaviors.

**Institutional Abbreviations**

**UWO-VPC** University of Wisconsin Oshkosh Vertebrate Paleontology Cast Collection, Oshkosh, WI, USA.

## ACKNOWLEDGEMENTS

We thank Collin Dischler (UWO) for assistance in producing tooth casts, Patty Ralrick, Mike Newbry (TMP), and James Farlow (IPFW), stimulating discussion and experimental design, Tom Suszek and Ben Sanderfoot (UWO) and Roy Plotnick (UIC) for assistance in running the flume systems. We also thank Heinrich Mallison (MfN) for encouragement and assistance at PeerJ. Special thanks to our PeerJ reviewers (Peter Dodson and Jordan Mallon), and editor (John Hutchinson) for helpful comments and suggestions.

### Funding

North-Central Section Geological Society of America Undergraduate Research Fund and the University of Wisconsin-Oshkosh Faculty/Student Collaborate Grant program provided financial support for this project. The funders had no role in study design, data collection and analysis, decision to publish, or preparation of the manuscript.

## Grant Disclosures

The following grant information was disclosed by the authors:

North-Central Section Geological Society of America Undergraduate Research Fund.

University of Wisconsin-Oshkosh Faculty/Student Collaborate Grant.

## Competing Interests

The authors declare there are no competing interests.

## Author Contributions

- Joseph E. Peterson conceived and designed the experiments, contributed reagents/materials/analysis tools, wrote the paper, prepared figures and/or tables, reviewed drafts of the paper.
- Jason J. Coenen conceived and designed the experiments, performed the experiments, wrote the paper, reviewed drafts of the paper.
- Christopher R. Noto analyzed the data, wrote the paper, prepared figures and/or tables, reviewed drafts of the paper.

## Data Deposition

The following information was supplied regarding the deposition of related data:

Figshare:

http://dx.doi.org/10.6084/m9.figshare.941093.

http://dx.doi.org/10.6084/m9.figshare.941092.

http://dx.doi.org/10.6084/m9.figshare.941095.

http://dx.doi.org/10.6084/m9.figshare.941094.

## Supplemental information

Supplemental information for this article can be found online at http://dx.doi.org/10.7717/peerj.347.

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
