# Peer review of "Fluvial transport potential of shed and root-bearing dinosaur teeth from the late Jurassic Morrison Formation"

_PeerJ, doi:10.7717/peerj.347_

## Round 0.1 · original submission · Minor Revisions

We've obtained constructive reviews from Dodson and Mallon that make some excellent points for future studies as well as for revising the paper. I agree that Mallon's points 3,4,6 are really important to address in the MS to maximize the value and clarity of the study; his other points deserve incorporation, too. Dodson also raises some points to consider and address in the Discussion.

Please ensure you include a point-by-point Response when submitting the new version. If I am satisfied by this, I'll accept the revised paper without further review.

·

Basic reporting

Satisfactory in all fundamentals except as noted.

Experimental design

Satisfactory in part. The submission would be improved if more experimental work were performed with modern teeth and tooth casts, and also with a variety of substrates. Hopefully such experiments will be done in future.

Validity of the findings

The most important finding is that teeth are more likely to be transported when they are shed rather than rooted. MOre is not claimed than the evidence suggests.

Additional comments

This report describes a set of experiments investigating transport factors in several dinosaur teeth. An important finding is that shed teeth are more mobile than teeth with roots. The study is interesting and publishable, but it also raises a series of questions. Teeth are among the densest of biological materials. The use of lightweight casts as proxies is the first thing to question. It may be justifiable inasmuch as fossil teeth are likely to be more dense than their living equivalents, but one can imagine a parallel set of calibration experiments that involve making casts of teeth of living animal and then using the natural teeth as controls. It would be good to use a graded series of mammalian teeth. Does it make any sense to manipulate the density of the casts to approximate those of natural teeth? The other concern I have is smooth glassy substrate. I would imagine that in various natural substrates that burial of a tooth in situ could be the result. Again, if the substrate were coarse and the tooth relatively small that the tooth could be invisible to the passing current. I thus can imagine many ways to extend the experimental work to obtain a broader and more useful picture of tooth taphonomy.
.

l. 117 – 119: elongate and conical are shapes, not sizes or colors. It is redundant to say “conical-shaped.” On the other hand, compact is not a shape. Just say compact!

l. 125: what does “distal proximity” mean?

l. 216: caps please!

l. 255: why relative distances? 50 cm is an absolute measure not a relative one.

Fig. 5: I recommend adding label as in Gif. 4 of parallel and perpendicular to A and B to make it clear how the two charts differ.

·

Basic reporting

The paper is concise, well-presented, and clearly written. It is a good start to what will ultimately be an interesting and useful addition to the taphonomy literature. I feel there is much work yet to be done, however, including the following concerns:
(1) The paper should be framed more explicitly as an hypothesis test.
(2) The first paragraph of the Discussion repeats the results and should be re-written.
(3) More problematically, I feel as though the manuscript is only half-baked. The authors do a fine job of demonstrating the differential transportability of root-bearing and shed teeth. However, I feel as though the palaeoecological ramifications of this observation were not fully considered. To what extent, if any, can these differences in transportability be corrected for? How do these observations impact real-world palaeoecological interpretations of microsites (e.g., Quarry 9 of the Morrison Formation)? Such additional analyses would greatly improve the paper and enhance its impact. Without them, the paper highlights an important taphonomic problem, but offers no help otherwise.
(4) I'm not certain that the authors' observations re: the association of root-bearing teeth with associated skeletons, and the association of shed with feeding sites, has any bearing whatsoever on the relative transportability of said teeth. For example, shed teeth (lacking roots) aren’t likely to occur near associated skeletons in the first place (unless they belong to a predator) because the teeth are shed in life, as the animal is on the move, not after death. Similarly, it isn't clear how the association of shed teeth with feeding sites relates to transportability. This needs further explaination.

Experimental design

The experiment, as presented, is well-executed. I do have some minor concerns, however:
(5) I'm confused as to why a cast was not made of a root-bearing Camarasaurus tooth (i.e., why was the root fabricated?). If such teeth are common enough to warrant consideration, why was one not cast?
(6) I feel that the authors did not do an adequate job of demonstrating that their data meet the assumptions of ANOVA (e.g., normal distribution). This should be done a priori (e.g., using a Shapiro-Wilk test).

Validity of the findings

The findings are valid, but I wonder just how significant they are. In my experience, Late Cretaceous microsites in Alberta uniformly yield relatively few teeth of any kind with intact roots. Given this low variability, can such rare root-bearing teeth be said to have any taphonomic/palaeoecologic significance? My suspicion is 'not really'. Again, the consideration of a real-world example would really help to drive home the importance of this paper.

Additional comments

See ms mark-up for additional (minor) comments.

---

## Round 0.2 · accepted · Accept

I have checked the rebuttal with the revision and am satisfied that the reviewers' comments have been taken seriously, and no further revisions are needed. This paper is now accepted- congrats!